# Associations between Health-Related Physical Fitness and Cardiovascular Disease Risk Factors in Overweight and Obese University Staff

**DOI:** 10.3390/ijerph17239031

**Published:** 2020-12-03

**Authors:** Jiangang Chen, Yuan Zhou, Xinliang Pan, Xiaolong Li, Jiamin Long, Hui Zhang, Jing Zhang

**Affiliations:** 1Department of Exercise Science, School of Physical Education, Shaanxi Normal University, Xi‘an 710119, China; chenjiangang@snnu.edu.cn (J.C.); yuanzhou@snnu.edu.cn (Y.Z.); lixiaolong@snnu.edu.cn (X.L.); longjiamin@snnu.edu.cn (J.L.); 41710161@snnu.edu.cn (H.Z.); 2School of Kinesiology, Beijing Sport University, Beijing 100084, China; panxl@snnu.edu.cn

**Keywords:** university staff, health-related physical fitness, cardiovascular disease, overweight, obesity

## Abstract

Purpose: This cross-sectional study examined the associations between health-related physical fitness (HPF) and cardiovascular disease (CVD) risk factors in overweight and obese university staff. Methods: A total of 340 university staff (109 women, mean age 43.1 ± 9.7 years) with overweight (*n* = 284) and obesity (*n* = 56) were included. The HPF indicators included skeletal muscle mass index (SMI), body fat percentage (BFP), grip strength (GS), sit-and-reach test (SRT), and vital capacity index (VCI). CVD risk factors were measured, including uric acid (UA), triglycerides (TG), high-density lipoprotein cholesterol (HDL-C), low-density lipoprotein cholesterol (LDL-C), and glucose (GLU). Results: BFP, SMI, and GS were positively associated with UA level (*β* = 0.239, *β* = 0.159, *β* = 0.139, *p* < 0.05). BFP was positively associated with TG and TG/HDL-C levels (*β* = 0.421, *β* = 0.259, *p* < 0.05). GS was positively associated with HDL-C level (*β* = 0.244, *p* < 0.05). SRT was negatively associated with GLU level (*β* = −0.130, *p* < 0.05). Conclusions: In overweight and obese university staff, body composition, muscle strength, and flexibility were associated with CVD risk factors. An HPF test may be a practical nonmedical method to assess CVD risk.

## 1. Introduction

Cardiovascular disease (CVD) is a significant public health issue, as it is the leading cause of adult mortality, accounting for more than 40% of deaths in China [1]. Over the past 30 years, the number of CVD deaths in China has increased from 2.51 million to 3.97 million annually [2]. Although the age-standardized mortality rate remained stable overall from 2002 to 2016, it increased among the young population [3]. Therefore, the early prevention of CVD is essential [4].

The risk factors of CVD include dyslipidemia, diabetes, and obesity [5]. In recent years, hyperuricemia has also been recognized as a potential risk factor for CVD, following the discovery of a causal relationship between uric acid and the adverse outcomes of CVD [6,7]. Among the many risk factors, obesity affects cardiovascular disease in several ways; for instance, obesity affects the morbidity of CVD, and early obesity may increase the risk of future CVD events [8]. Obesity also affects the prognosis of CVD. A meta-analysis showed that overweight and obese individuals had 25% and 42% increased risks of CVD mortality, respectively, compared to those of normal-weight individuals [9]. Furthermore, the duration of obesity may also affect CVD. Abdullah et al. [10] found that every two years of obese living significantly increased the risk of CVD mortality by 7%. This may be because obesity worsens other CVD risk factors such as blood lipid and blood glucose levels [11]. It is necessary, therefore, to assess the risk factors of CVD in overweight and obese individuals.

Common measures of obesity include body mass index (BMI) and body composition. Body composition is one of the components of health-related physical fitness (HPF). In addition to body composition, other HPF components include cardiorespiratory fitness, muscular fitness, and flexibility. HPF reflects not only the ability of the body to participate in exercise, but also the body’s ability to reduce the risk of disease. HPF assessment, therefore, may have important implications in the prevention of chronic diseases. Previous studies have shown that HPF can predict multiple risk factors for CVD [12,13,14]. Body composition is better at distinguishing between fat mass and fat-free mass than BMI. Studies have shown that individuals with the same BMI may differ in body composition, which affects their risks of CVD [15]. Cardiorespiratory fitness (CRF) is one of the most important HPF factors, and it is a strong predictor of CVD [16,17]. The respiratory function, however, also plays an important role in cardiovascular health [18], and it is not clear whether indicators of respiratory function can be used to assess the risk of CVD. In recent years, the relationship between muscular fitness and CVD has gradually attracted attention. Grip strength has a negative correlation with triglyceride and glucose levels [19], and decreased respiratory muscle strength is an independent risk factor for CVD [20]. The relationship between muscle strength and uric acid level may vary between populations. One study reported a negative correlation between grip strength and uric acid in young people but a positive correlation in older people [21]. Flexibility may also be an indicator of CVD risk. Chang et al. found that flexibility was positively correlated with high-density lipoprotein levels among 628 community residents but did not control other variables such as sex and age [22]. Thus, the relationship between flexibility and CVD requires further exploration.

To the best of our knowledge, numerous studies have been conducted on the relationship between HPF and CVD in normal-weight individuals [13,22]. No study, however, has focused on overweight and obese university staff. A cross-sectional study found that 94 percent of university staff were exposed to one or more CVD risk factors [23]. This may indicate that university staff have a higher risk of cardiovascular disease. Furthermore, university staff have a higher prevalence of overweight and obesity than the general population does because of longer working hours and psychosocial factors [24]. Cardiovascular disease risk factors, such as blood lipids, glucose, and uric acid levels, may further worsen among university staff in overweight and obese states. The purpose of this study, therefore, was to explore the associations between HPF and CVD risk factors in overweight and obese university staff and to provide a basis for HPF testing in evaluating the risk of CVD and the development of future exercise intervention strategies.

## 2. Materials and Methods

### 2.1. Participants and Study Design

From October 2019 to January 2020, a total of 2800 university staff underwent annual health screenings at the Community Hospital Health Management Center. This cross-sectional study recruited university staff every morning in the breakfast serving area of the Health Management Center. A sample size of 319 was required to achieve 90% statistical power based on the calculation of PASS.11.0, and 412 university staff were actually recruited. Among the 412 participants, 38 participants did not complete HPF tests, and 34 participants were over 60 years. A total of 340 overweight and obese university staff aged between 25 and 60, with a BMI greater than 24.0, were eventually included in the study. The classification criteria for overweight and obesity were from the China Obesity Working Group [25].

Participants first underwent a blood collection procedure, and then height, weight, and body composition were measured. Other HPF tests were performed after participants ate breakfast and took a 15 min break to regain their strength. The participants were informed of the test procedures, requirements, and possible risks before the test, and they signed the informed consent forms. The protocols were approved by the Ethics Committee of Shaanxi Normal University, and the ethical approval code is 202016003.

### 2.2. Study Variables

#### 2.2.1. Health-Related Physical Fitness Measurement

Health-related physical fitness indicators were measured by trained research assistants according to the National Physical Fitness Standards Manual.

Body height and weight were measured using an all-in-one machine (GK 720, Shandong, China), which combined an ultrasonic stadiometer with an electronic weight scale. The machine was calibrated before each use to ensure accuracy. Participants stood barefoot in a designated position and looked straight ahead. The test results were automatically recorded and reported.

Body composition was assessed with a bioelectrical impedance machine (InBody 230, Seoul, South Korea). Participants stood barefoot on the electrodes of the machine and held the handles with both hands. Participants remained in a natural standing position throughout the test and always kept their hands and feet in contact with the electrodes. The test program took 1 to 2 min, and skeletal muscle mass (SMM) and total body fat mass (BFM) were automatically recorded by the InBody 230. Skeletal muscle mass index (SMI) was determined using SMM divided by height squared. Body fat percentage (BFP) was determined using BFM divided by body weight.

Grip strength was measured using a portable electronic grip strength dynamometer (Hengkangjiaye, Guangzhou, China). Participants stood naturally with their arms slanting down and their palms inward. Participants were not allowed to swing their arms or hold the dynamometer close to their bodies during the test. Each hand was tested twice, and the highest measurements were recorded.

Flexibility was assessed with the sit-and-reach test (SRT). An electronic fleximeter (HKD-1442, Beijing, China) was used to achieve better accuracy. Participants sat barefoot in the required position with their legs straight and their feet together. During each measurement, participants pushed their feet against the front baffle and stretched their arms as far forward as possible. Measurements were recorded to evaluate flexibility as participants tried to push the cursor on the farthest scale with the fingertips of their hands. Participants tried three times, and the largest measurements were recorded.

Vital capacity (VC) was measured using an electronic pneumometer (WCS-1000, Beijing, China). Participants inhaled deeply and then exhaled all the air to assess vital capacity. Participants tried this three times, and the maximum exhalation was recorded. Participants were asked to rest 30–60 s between the three measurements in order to avoid hypoxia. Vital capacity Index (VCI) was determined using VC divided by body weight.

#### 2.2.2. Cardiovascular Disease Factors Measurement

After a night of fasting, blood samples were collected from 8 a.m. to 10 a.m. in the Community Hospital Health Management Center. On the same day, blood specimens were processed and analyzed in the laboratory of the Community Hospital. The automatic analyzer AU480 (Beckman Coulter, California, USA) was used to analyze blood biochemistry. Serum uric acid (UA), triglycerides (TG), total cholesterol (TC), high-density lipoprotein cholesterol (HDL-C), and low-density lipoprotein cholesterol (LDL-C) were measured with enzymatic methods. Blood glucose (GLU) was measured using a hexokinase enzymatic method. TG/HDL-C was, thereafter, used to estimate insulin resistance [26].

### 2.3. Statistical Analysis

All variables were checked for normality using the Kolmogorov–Smirnov test. Continuous variables with normal distribution were presented by mean ± standard deviation (SD), continuous variables with non-normal distribution were presented by median (interquartile range (IQR)), and categorical variables were presented by number (percentage). Multiple linear regression was used to analyze the associations between HPF and CVD risk factors. Each variable was z-standardized before the regression analysis to compare these values on the same scale. A two-sided *p*-value < 0.05 was considered statistically significant. SPSS 23.0 software (Chicago, IL, USA) was used for statistical analyses.

## 3. Results

Table 1 illustrates the baseline characteristics of participants. A total of 340 university staff (109 women, 231 men) were included in this study. Participants’ mean age was 43.1 ± 9.7 years, and 9.1% of participants were 25–30 years, 35.6% were 31–40 years, 26.8% were 41–50 years, and 28.5% were 51–60 years old. Participants’ mean BMI was 26.2 ± 1.9 kg/m^2^, 83.5% of participants were overweight, and 16.5% were obese.

Table 2 shows the associations between HPF indicators and CVD risk factors. BFP, SMI, and GS were positively associated with UA level (*β* = 0.239, *β* = 0.159, *β* = 0.139, *p* < 0.05). BFP was positively associated with TG and TG/HDL-C levels (*β* = 0.421, *β* = 0.259, *p* < 0.05). GS was positively associated with HDL-C level (*β* =0.244, *p* < 0.05). SRT was negatively associated with GLU level (*β* = −0.130, *p* < 0.05).

## 4. Discussion

This study evaluated the associations between HPF indicators and CVD risk factors in overweight and obese university staff. The main findings of this study were that reduced flexibility was associated with elevated GLU level, while high body fat percentage, muscle mass, and grip strength were associated with high UA level. We also observed that grip strength was positively associated with high-density lipoprotein cholesterol (HDL-C) level and that body fat percentage was positively associated with TG and TG/HDL-C levels. These results indicated that the HPF and CVD risk factors were related, and they provide a basis for nonmedical evaluations of CVD risk in overweight or obese university staff and the development of future exercise intervention strategies.

Our analysis of the association between HPF indicators and UA showed that body fat percentage was positively associated with UA level. This may be attributed to the fact that purine metabolism in adipose tissue is enhanced in obesity [27,28]. Furthermore, the distribution of body fat is closely related to UA level [29]. Huang et al. reported that visceral fat accumulation increased the risk of hyperuricemia in older Chinese adults [30]. This finding suggests that, in addition to the total body fat percentage, visceral fat is also an important indicator to consider in the prevention of hyperuricemia. We also observed that skeletal muscle index and grip strength were positively associated with UA level. This finding is consistent with those of previous cross-sectional studies in the elderly [31,32,33]. UA is the final product of purine metabolism. An excessive accumulation of UA in the body may cause not only gout but also heart failure [6,7]. In recent years, UA has been observed to slow age-related muscle decline [33,34]. In a longitudinal study, Macchi et al. [35] found that in people with an average age of 76 years, higher baseline serum UA levels were associated with better muscle function three years later. This muscular protection, however, was not observed in those under 60 years of age [36]. Furthermore, the possible physiological mechanisms by which UA protects muscles are not clear. Although studies have assumed that UA plays a protective role in the process of free radical damage to skeletal muscle protein [37], UA is also a pro-oxidant and may increase oxidative stress. Another study suggests that UA may be an indicator of dietary protein intake. High UA concentrations in patients with hyperuricemia are associated with better nutritional status [38]. Total protein intake, particularly those of meat and fish proteins, may be important for building and maintaining muscle mass [39]. Individuals with higher dietary protein intake, therefore, may maintain higher muscle mass as well as higher UA levels. Previous studies have shown that high body fat and muscle mass both place a burden on the cardiovascular system and increase the risk of cardiovascular disease [11], indicating that weight control and improvement to body composition are important for overweight and obese university staff.

Our study results revealed that flexibility was negatively associated with GLU level. Aparicio et al. [40] also reported a negative correlation between flexibility and GLU in menopausal women; however, their finding was not statistically significant, probably because flexibility was a self-reported scale score rather than an actual measured value. One possible explanation for the association between flexibility and GLU is disc degeneration. Hyperglycemia has a detrimental effect on disc cell viability, leading to disc degeneration and impaired lumbar flexibility [41]. Inflammatory cytokines, which mediate insulin resistance, also play a role in disc degeneration [42,43]. The relationship between flexibility and GLU metabolism, however, remains poorly understood, and as many factors affecting flexibility and blood glucose are not considered, the exact mechanism is not clear. Studies in recent years have begun to recognize the value of flexibility in evaluating and preventing chronic diseases. Gregorio et al. [44] reported that flexibility not only in the waist but also in the upper body is associated with cardiometabolic risk factors. Another study found that flexibility exercises reduced pro-inflammatory adipokines, such as PAI-1 and chemerin, and increased anti-inflammatory adipokines, such as adiponectin [45]. Future studies are needed to further confirm the effectiveness of flexibility training in reducing blood glucose and preventing CVD.

Our analysis of the relationship between HPF and blood lipids revealed that body fat percentage was positively associated with TG and TG/HDL-C levels. This finding is supported by previous studies using dual-energy X-rays and confirmed that body fat and its distribution are closely related to blood lipid levels [46,47]. Konieczna et al. [46] further compared regional and total body fat measurements, reporting that the ratio of visceral adipose tissue to total fat was a more effective evaluation indicator of TG. This may be because visceral fat mediates partial insulin resistance through the release of inflammatory adipokines [46]. As a result, lipolysis is intensified, and excess TGs enter the liver, causing an abnormally high TG level [48].

We observed that grip strength was positively associated with HDL-C level. Grip strength is an effective indicator of muscle strength and of potential health risks [49]. In a study of 8576 participants, Lee et al. found that participants with low grip strength had increased risks of CVD [50]. Another large sample study determined that higher relative grip strength was associated with healthier blood lipid levels in adults, such as lower TG and total cholesterol levels and higher HDL-C levels [19]. This may be related to the endocrine function of skeletal muscle and its metabolic benefits [51]. Cytokines secreted by the muscles may regulate the metabolic process through autocrine and paracrine mechanisms. Cytokines, such as myonectin and irisin, regulate lipid metabolism and improve insulin resistance [52,53].

One advantage of this study was that we first explored the associations between HPF and CVD risk factors in overweight or obese adults and provided new insight regarding CVD prevention and control strategies. Secondly, the HPF indicators included in this study were easy to measure and obtain. It is convenient, therefore, for the general population to conduct HPF self-assessments. Certainly, this study also has some limitations. Firstly, physical activity, nutritional status, and physiological indicators related to inflammation, such as blood pressure and CRP, were not included in the study. Secondly, our study cannot determine the causal relationship between HPF and CVD risk factors due to the cross-sectional design. Thirdly, the results of this study cannot be generalized for the overall population, as participants were overweight and obese adults. In the future, longitudinal or experimental studies should be considered to verify the causal relationship between HPF and CVD risk factors.

## 5. Conclusions

Among overweight and obese university staff, reduced flexibility was associated with high glucose level, while high body fat percentage, muscle mass, and grip strength were associated with high uric acid level. Additionally, grip strength was positively associated with HDL-C level, and body fat percentage was positively associated with TG and TG/HDL-C levels. The results of this study suggest that body composition, grip strength, and flexibility may be practical nonmedical markers for assessing cardiovascular disease risk. In the future, prospective studies should be conducted to investigate the extent to which exercise programs that improve body composition and increase muscle strength and flexibility may reduce the risk of cardiovascular disease.

## Figures and Tables

**Table 1 ijerph-17-09031-t001:** Baseline characteristics of the participants.

	Men	Women	All
Sample size (*n*, %)	231 (67.9%)	109 (32.1%)	340 (100%)
Age (*n*, %)			
≤30 years	20 (8.7%)	11 (10.1%)	31 (9.1%)
31–40 years	80 (34.6%)	41 (37.6%)	121 (35.6%)
41–50 years	60 (26.0%)	31 (28.4%)	91 (26.8%)
51–60 years	71 (30.7%)	26 (23.9%)	97 (28.5%)
BMI (*n*, %)			
24–27.9 (overweight)	192 (83.1%)	92 (84.4%)	284 (83.5%)
≥ 28.0 (obese)	39 (16.9%)	17 (15.6%)	56 (16.5%)
HPF indicators			
Skeletal muscle mass (kg) ^a^	33.07 ± 2.86	24.40 ± 2.49	30.29 ± 4.86
Skeletal muscle mass index (kg/m^2^) ^a^	11.00 ± 0.62	9.17 ± 0.68	10.43 ± 1.06
Body fat mass (kg) ^b^	20.00 (5.80)	24.10 (5.00)	21.20 (6.40)
Body fat percentage (%)^b^	25.00 (5.06)	35.55 (4.34)	27.48 (9.64)
Grip strength (kg) ^b^	36.50 (9.40)	25.60 (5.20)	33.60 (11.10)
Sit-and-reach (cm) ^b^	4.10 (11.50)	9.15 (11.80)	5.90 (11.00)
Vital capacity (mL) ^a^	3957.94 ± 844.10	2708.75 ± 624.92	3564.65 ± 979.17
Vital capacity index (mL/kg) ^a^	50.66 ± 10.59	39.71 ± 9.54	47.40 ± 11.44
CVD risk factors			
UA (umol/L) ^b^	384.00 (72.00)	301.50 (65.00)	363.00 (97.00)
TG (mmol/L) ^b^	1.35 (0.79)	1.10 (0.65)	1.27 (0.79)
HDL-C (mmol/L) ^a^	1.32 ± 0.24	1.54 ± 0.25	1.38 ± 0.42
LDL-C (mmol/L) ^b^	2.96 (0.74)	2.65 (0.91)	2.92 (0.86)
TG/HDL-C ratio ^b^	1.01 (0.70)	0.70 (0.53)	0.91 (0.72)
GLU (mmol/L) ^a^	5.03 ± 0.43	5.07 ± 0.42	5.05 ± 0.51

Note: ^a^ Data are represented by mean ± SD; ^b^ Data are represented by median (IQR). Abbreviations: BMI: body mass index; UA: uric acid; TG: triglycerides; HDL-C: high-density lipoprotein cholesterol; LDL-C: low-density lipoprotein cholesterol; GLU: blood glucose.

**Table 2 ijerph-17-09031-t002:** Associations between health-related physical fitness (HPF) indicators and cardiovascular disease (CVD) risk factors.

Dependent Variables ^a^	Independent Variables ^a^	*β*	*β* (95%CI)	SE	*p*	R^2^
	SMI	0.159	(0.001, 0.318)	0.081	0.049 *	
	BFP	0.239	(0.076, 0.402)	0.083	0.004 *	
UA	GS	0.139	(0.018, 0.259)	0.061	0.024 *	0.363
	SRT	0.027	(−0.068, 0.122)	0.048	0.579	
	VCI	0.031	(−0.086, 0.147)	0.059	0.608	
	SMI	0.162	(−0.030, 0.353)	0.097	0.098	
	BFP	0.421	(0.226, 0.617)	0.099	0.000 *	
TG	GS	0.031	(−0.113, 0.175)	0.073	0.673	0.098
	SRT	−0.036	(−0.149, 0.078)	0.058	0.538	
	VCI	0.047	(−0.092, 0.187)	0.071	0.507	
	SMI	−0.183	(−0.370, 0.003)	0.095	0.054	
	BFP	0.014	(−0.177, 0.205)	0.097	0.887	
HDL-C	GS	0.244	(0.103, 0.385)	0.072	0.001 *	0.128
	SRT	−0.009	(−0.121, 0.103)	0.057	0.871	
	VCI	0.066	(−0.072, 0.203)	0.070	0.349	
	SMI	0.045	(−0.148, 0.238)	0.054	0.646	
	BFP	0.131	(−0.068, 0.330)	0.010	0.197	
LDL-C	GS	−0.009	(−0.155, 0.136)	0.005	0.900	0.063
	SRT	0.097	(−0.019, 0.212)	0.004	0.102	
	VCI	0.004	(−0.138, 0.147)	0.004	0.954	
	SMI	0.150	(−0.043, 0.344)	0.099	0.128	
	BFP	0.259	(0.061, 0.457)	0.101	0.011 *	
TG/HDL-C	GS	−0.054	(−0.201, 0.092)	0.074	0.465	0.070
	SRT	−0.029	(−0.145, 0.087)	0.059	0.622	
	VCI	−0.008	(−0.151, 0.134)	0.072	0.909	
	SMI	0.181	(−0.007, 0.369)	0.096	0.059	
	BFP	0.128	(−0.065, 0.321)	0.098	0.192	
GLU	GS	0.035	(−0.107, 0.177)	0.072	0.630	0.083
	SRT	−0.130	(−0.243, −0.017)	0.057	0.024 *	
	VCI	−0.052	(−0.190, 0.086)	0.070	0.461	

Note: the multiple linear regression model controlled for sex and age; ^a^ indicates the variable was z-standardized; * indicates statistical significance (*p* < 0.05); SE indicates standard error. Abbreviations: UA: uric acid; TG: triglycerides; HDL-C: high-density lipoprotein cholesterol; LDL-C: low-density lipoprotein cholesterol; GLU: blood glucose; SMI: skeletal muscle index; BFP: body fat percentage; GS: grip strength; SRT: sit-and-reach test; VCI: vital capacity index.

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
