# Peer review of "Associations between Health-Related Physical Fitness and Cardiovascular Disease Risk Factors in Overweight and Obese University Staff"

_ijerph, 2020, doi:10.3390/ijerph17239031_

Round 1

Reviewer 1 Report

Specific comments:

  1. The authors aimed to investigate the associations between health-related physical fitness and cardiovascular disease risk factors in overweight and obese university staff members in China. The one main limitation from this is that no normal weight staff were included to test these same associations in a “healthy” comparative group of staff members. Why did this study not include normal weight adults too?
  2. In addition, CVD risk factors used in this study included uric acid, lipids and glucose. This selection seems very strange especially since uric acid receives criticism in CVD related studies, because it reflects disorders related to purine metabolism. Also, no marker of inflammation such as CRP or blood pressure was included which is very important in the setting of overweight and obesity.
  3. It is not clear from the motivation of this study why men and women were analysed separately. This is especially a concern when women were only 32% of the study population. Only 56 participants (16.5%) were obese, how many of them are women and how many of them are men? Same for the overweight group? These values are not provided. Power analysis is required to determine the effect size (for obesity and sex) and to determine the significance level at an alpha probability <0.05. The rationale for sex differences is not clear in this study.
  4. The findings reported in the abstract does not make sense. There are beta coefficients indicated for one set of results, e.g. “In men, SMI and grip strength were positively associated with UA level (β = 16.955, β = 1.158, p < 0.05),…” Why are there two beta values? Why are the beta values so different? Are these unstandardised or standardised beta values? Have the appropriate z-scores been determined prior multivariable adjusted analyses in order to have all variables on the same scale for comparison?
  5. The authors wrote in the Introduction, lines 35-37: “ Although the age-standardized mortality rate remained stable overall from 2002 to 2016, it increased among the young population [3].” Saying this and presenting age categories in the study population, an age stratification would have been a better approach to indicate whether the health related physical fitness tests are appropriate in all age groups for the evaluation of CVD risk.
  6. The description of Table 2 does not fit the content of Table 2. There is “n” and not sure what this reflects? Why are the values not separately shown for men and women as mentioned in the text? It would be more relevant to show the differences in these variables between overweight and obese individuals and not men and women.
  7. It is not clear how the covariates were handles prior including them into the regression analyses. The beta values show large variance and the authors should indicate how this was managed. Each variable should be z-standardised in order to interpret these values on the same scale. Again, these associations may mask both type I and II errors due to the small number of obese individuals and that the n-value of men and women could be too small with a lack of power. This should be addressed. Also, nor adjusted R squared values were shown. What was the validity of each model?

Technical comments:

  1. Abbreviations are inconsistent. E.g. in the abstract vital capacity index is abbreviated as VPI and later as VCI.
  2. In the conclusion the authors should tone down the meaning of their findings, i.e. muscle mass and strength were “potent” predictors. Is this true? Can the authors predict with cross-sectional analysis?

Author Response

Response to Reviewer 1 Comments

Point 1: The authors aimed to investigate the associations between health-related physical fitness and cardiovascular disease risk factors in overweight and obese university staff members in China. The one main limitation from this is that no normal weight staff were included to test these same associations in a “healthy” comparative group of staff members. Why did this study not include normal weight adults too?

Response 1: Thanks for the comment. Your opinions are very valuable and helpful for us to make the introduction section more logical. We have added some new descriptions to make our main purpose more prominent (Page 2, Line 70-77). We have also added the corresponding description in the limitation section to indicate that our results are not suitable for extrapolating to other populations (Page 7, Line 237).

In fact, most previous studies have focused on people of normal weight. In these groups,  health-related physical fitness (HPF) has been found to be associated with risk factors of cardiovascular disease (CVD). Normal-weight university staff should also be part of the normal-weight population, so we did not compare these associations in the article. Our primary goal was to explore whether these associations also existed among overweight and obese staff.

In the next study we will certainly consider comparing these associations among staff of different BMI. Thank you very much for your comments.

Point 2:  In addition, CVD risk factors used in this study included uric acid, lipids and glucose. This selection seems very strange especially since uric acid receives criticism in CVD related studies, because it reflects disorders related to purine metabolism. Also, no marker of inflammation such as CRP or blood pressure was included which is very important in the setting of overweight and obesity.

Response 2: Thank you for the comment. We admitted that uric acid has been criticized in some previous studies as a risk factor for cardiovascular disease. However, recent studies have shown a causal relationship between uric acid and many cardiovascular events1-3. Thanks for your reminder. We have given a special explanation of the CVD risk of uric acid (Page, Line 37).

In the Limitations section, we have added that an important marker of inflammation for overweight and obesity was not included in this study (Page 7, Line 234).

1  Kleber, M.E.; Delgado, G.; Grammer, T.B.; Silbernagel, G.; Huang, J.; Kramer, B.K.; Ritz, E.; Marz, W. Uric Acid and Cardiovascular Events: A Mendelian Randomization Study. J. Am. Soc. Nephrol. 2015, 26, 2831–2838.

Chiang, K.M.; Tsay, Y.C.; Vincent, N.T.; Yang, H.C.; Huang, Y.T.; Chen, C.H.; Pan, W.H. Is Hyperuricemia, an 2  Early-Onset Metabolic Disorder, Causally Associated with Cardiovascular Disease Events in Han Chinese? J Clin Med. 2019, 8, 1202.

3 Ndrepepa G. Uric acid and cardiovascular disease. Clin Chim Acta. 2018 Sep;484:150-163. doi: 10.1016/j.cca.2018.05.046. Epub 2018 May 24. PMID: 29803897.

Point 3:  It is not clear from the motivation of this study why men and women were analysed separately. This is especially a concern when women were only 32% of the study population. Only 56 participants (16.5%) were obese, how many of them are women and how many of them are men? Same for the overweight group? These values are not provided. Power analysis is required to determine the effect size (for obesity and sex) and to determine the significance level at an alpha probability <0.05. The rationale for sex differences is not clear in this study.

Response 3: Thanks for the comment. We have unified table 1 and 2 to make our table clearer. The information you requested is fully displayed in the new table (Page 4, Line 148-151).

According to PASS11.0's calculation, the statistical power of our total sample size did indeed reach 0.92, the statistical power of the male sample also reached 0.82, while the power of the female sample was only about 0.5.

Therefore, we combined the sample of men and women, so that the problems you mentioned were all resolved together. Power analysis were included in the Study Design (Page , Line 84).

The differences between men and women may be due to sex hormones, body structure, lifestyle, and many other factors.

Point 4: The findings reported in the abstract does not make sense. There are beta coefficients indicated for one set of results, e.g. “In men, SMI and grip strength were positively associated with UA level (β = 16.955, β = 1.158, p < 0.05),…” Why are there two beta values? Why are the beta values so different? Are these unstandardised or standardised beta values? Have the appropriate z-scores been determined prior multivariable adjusted analyses in order to have all variables on the same scale for comparison?

Response 4: Thank you for the comment.  It is possible that you did not understand the abstract due to our inadequate expression. Since we combined the sample of men and women so that the presentation of the results changed accordingly. We believe that the problems you mentioned were all resolved.

According to your suggestion, each variable was z-standardized before the regression analysis to compare these values on the same scale (Page 3, Line 139).

Thank you again for your comments, which are very helpful for us to improve the manuscript.

Point 5: The authors wrote in the Introduction, lines 35-37: “ Although the age-standardized mortality rate remained stable overall from 2002 to 2016, it increased among the young population [3].” Saying this and presenting age categories in the study population, an age stratification would have been a better approach to indicate whether the health related physical fitness tests are appropriate in all age groups for the evaluation of CVD risk.

Response 5: Thanks for the comment. Your opinions are of great help to our scientific research work. In our next study, we will try to use age stratification for analysis. In this study, however, only a limited sample of overweight and obese staff. As a result, age stratification raises the issue of statistical power, which is the same problem when analyzed men and women separately.

Point 6: The description of Table 2 does not fit the content of Table 2. There is “n” and not sure what this reflects? Why are the values not separately shown for men and women as mentioned in the text? It would be more relevant to show the differences in these variables between overweight and obese individuals and not men and women.

Response 6: Thank you for the comment. We admitted gender should be indicated instead of sample size (n) and percentage (%).  We have unified table 1 and 2 to make our table clearer (Page 4, Line 148-151). In our study, overweight and obese people were defined as those with a BMI greater than 24. Our main purpose is not to distinguish them, but to see them as a whole. And because the sample size of obese people is so small, it is not appropriate to compare the differences between them. In the future research, we will try the research ideas you mentioned. Thank you very much.

Point 7: It is not clear how the covariates were handles prior including them into the regression analyses. The beta values show large variance and the authors should indicate how this was managed. Each variable should be z-standardised in order to interpret these values on the same scale. Again, these associations may mask both type I and II errors due to the small number of obese individuals and that the n-value of men and women could be too small with a lack of power. This should be addressed. Also, nor adjusted R squared values were shown. What was the validity of each model?

Response 7: Thanks for the comment. In the new manuscript, important information such as R2 has been fully presented. We have modified the regression analysis table (Page 5, Line 158-164).

According to your suggestion, each variable was z-standardized before the regression analysis to compare these values on the same scale. Therefore, these beta values is the standardized beta values.

Since we combined the sample of men and women, the presentation of the results changed accordingly, so that the problems you mentioned were all resolved together.

Point 8: Abbreviations are inconsistent. E.g. in the abstract vital capacity index is abbreviated as VPI and later as VCI.

Response 8: Thank you for the comment. We have been aware of the mistake and corrected it in the manuscript. According to previous research and grammar rules, the abbreviation of vital capacity should be VC (Page 3, Line 122), the abbreviation of vital capacity index should be VCI.

Point 9: In the conclusion the authors should tone down the meaning of their findings, i.e. muscle mass and strength were “potent” predictors. Is this true? Can the authors predict with cross-sectional analysis?

Response 9: Thanks for the comment. Your opinion is very important for us to correct the mistakes in the manuscript. Scientific papers should be carefully and objectively worded, and we recognize that some powerful words should not be used. We re-examined all parts of the manuscript and corrected them accordingly (Page 7, Line 245).

Reviewer 2 Report

The paper by Chen et al is aimed at assessing the relationship between health-related physical fitness and several cardiovascular disease risk factors in overweight and obese university staff.

The topic may be interesting to Know how different components of health-related physical fitness have associated to cardiovascular disease risk factors in a sample of Chinese adults. However, the paper has several methodological concerns that should be resolved.

Major Concerns

Title and abstract: The author`s should Indicate the study’s design in the title or the abstract

Introduction:

Line 53.  The term health-related physical fitness (HPF) and its components should be clarified

Line 55: Although the authors mention the Cardiorespiratory fitness as one of the most important HPF factors with an important effect on cardiovascular health, and it has been generally stated that is a strong and independent predictor of cardiovascular disease and all-cause mortality, this Cardiorespiratory Fitness hasn’t been measured in the study (the consultation of this article can be helpful. https://www.who.int/bulletin/volumes/96/11/18-213728/en/). Vital capacity index is usually used to assess respiratory function. The importance of vital capacity in relation with cardiovascular health should be described.

Methods

Line 75: the study design should be described.

Line 76: The period of recruiting should be reported. The location of the study should be mentioned. How many people underwent the Health Screaning?

This is a sample of Chinese adults who are university staff. Has this specific occupation more risk to develop cardiovascular disease? If so, it should be justified in the introduction.

Sample size: How was calculated the sample sized? This section should be added.

Line 90: How many measures were taken?

Statistical method

Line 141: Table 2 is confusing, because the authors try to show the HPF indicators and CVD risk factors in different sex groups, but the sex is not represented in the table. I recommend you to unify table 1 and 2. This new table should show the descriptive characteristics of the study sample by sex. In tables, it is common to add the p value to distinguish between p<0.05 and p<0.001. The parametric data analysis represented by “a” and the non parametric analysis represented by “b” should be written in each variable.

Line 154: SE acronym should be explained in the table footnote

Discussion

Line 168,169, 188, 204: the authors use the term “correlation” instead of “association”. I don’t know if it is a mistake with the English term. If so, you should check throughout the text to amend this error.

If the correlation analysis has been done (Pearson or Spearman correlation analysis), it should be added in the result section adding a new table.

Line 175: the authors affirm that “UA in the body causes not only gout but also CVD such as hypertension and heart failure”. Hypertension is generally accepted to be a cardiovascular risk factor not a cardiovascular disease. This concept should be clarified.

Line 188: the authors discuss the relationship between body fat percentage and UA level in female staff. This finding has not been reported in the summarized key results (Line 164-169). It should be included.

169-170: The authors affirm that “These results indicated that the HPF test can be used as a predictor of UA, blood lipid, and GLU levels for overweight and obese individuals”. As the associations between HPF and Glucose is only significant for women, this affirmation should be modified because individuals are men and women.

Line 241: Vital capacity is only significant in women. This fact should be mentioned

Line 239: The authors affirm: “As one of the indicators for evaluating CRF, vital capacity…”. The concept of vital capacity as part of CRF should be reviewed.

Limitations: It should be included that the study results cannot be extrapolated to the overall population because the sample only includes adults who were university staff.

Minor Concerns

Line 82. Before this line, I recommend to add a section entitled Study Variables

Line 174: I suggest to Hypothesised with “may or might” before the verb “cause”

Line 189: I suggest to change the preposition “during”

Author Response

Response to Reviewer 2 Comments

Point 1: Title and abstract:  The author`s should Indicate the study’s design in the title or the abstract.

Response 1: Thanks for the comment. We added "cross-sectional study" in the first sentence of the abstract, indicating the type of study design (Page 1, Line 13).

Point 2:  Line 53.  The term health-related physical fitness (HPF) and its components should be clarified

Response 2: Thank you for the comment. We have added a description of health-related physical fitness (HPF) and its components in the introduction  (Page 2, Line 47,48,49).

Point 3:  Line 55: Although the authors mention the Cardiorespiratory fitness as one of the most important HPF factors with an important effect on cardiovascular health, and it has been generally stated that is a strong and independent predictor of cardiovascular disease and all-cause mortality, this Cardiorespiratory Fitness hasn’t been measured in the study (the consultation of this article can be helpful. https://www.who.int/bulletin/volumes/96/11/18-213728/en/). Vital capacity index is usually used to assess respiratory function. The importance of vital capacity in relation with cardiovascular health should be described.

Response 3: Thanks for the comment. We have added information in the introduction to indicate that vital capacity is important for cardiovascular health  (Page 2, Line 57).

Point 4: Line75 the study design should be described.

Response 4: Thank you for the comment. We have described the study design according to your suggestion  (Page , Line 82).

Point 5: Line 76: The period of recruiting should be reported. The location of the study should be mentioned. How many people underwent the Health Screaning?This is a sample of Chinese adults who are university staff. Has this specific occupation more risk to develop cardiovascular disease? If so, it should be justified in the introduction.

Response 5: Thanks for the comment. We added the time and place of recruitment to the Participants section and provided information on the total number of participants who underwent the Health Screening (Page 2, Line 80). In the introduction, we added information to clarify that university staff are at greater risk of cardiovascular disease (Page , Line71,72 ). Thank you very much for this suggestion, which is very helpful to us.

Point 6: Sample size: How was calculated the sample sized? This section should be added.

Response 6: Thanks for the comment. To achieve a statistical power of 90% of our study, we combined male and female samples. The sample size was calculated based on a previous study [1] and PASS 11.0. We have added the corresponding description in the Study Design section.  (Page , Line 83, 84).

[1] Norman G, Monteiro S, Salama S. Sample size calculations: should the emperor's clothes be off the peg or made to measure? BMJ. 2012 Aug 23;345:e5278. doi: 10.1136/bmj.e5278. Erratum in: BMJ. 2014;349:g5341. PMID: 22918496.

Point 7: Line 90: How many measures were taken?

Response 7: Thank you for the comment. Height and weight were measured only once. Since we use an all-in-one machine, the measurements of height and weight are automatically recorded and reported, so there is no deviation caused by manual operation. The results of the one test are reliable. More importantly, the machine is calibrated before each test, and this information has been added to the manuscript (Page 3, Line 100).

Point 8: Line 141: Table 2 is confusing, because the authors try to show the HPF indicators and CVD risk factors in different sex groups, but the sex is not represented in the table. I recommend you to unify table 1 and 2. This new table should show the descriptive characteristics of the study sample by sex. In tables, it is common to add the p value to distinguish between p<0.05 and p<0.001. The parametric data analysis represented by “a” and the non parametric analysis represented by “b” should be written in each variable.

Response 8: Thank you for comments. We admitted that two typos did occur in table 2 and that gender should be indicated instead of sample size (n) and percentage (%). We have a unified table 1 and 2. We can't agree with your suggestion more because it does make our table clearer (Page 4, Line 147-151).

Point 9: Line 154: SE acronym should be explained in the table footnote

Response 9: Thank you very much for giving us such valuable advice. We have added the corresponding explanation in the table footnote (Page 5, Line 160).

Point 10: Line 168,169, 188, 204: the authors use the term “correlation” instead of “association”. I don’t know if it is a mistake with the English term. If so, you should check throughout the text to amend this error.

If the correlation analysis has been done (Pearson or Spearman correlation analysis), it should be added in the result section adding a new table.

Response 10: Thank you very much for giving us such valuable advice. We have corrected the errors in the whole article to ensure that the regression analysis is expressed correctly.

Point 11: Line 175: the authors affirm that “UA in the body causes not only gout but also CVD such as hypertension and heart failure”. Hypertension is generally accepted to be a cardiovascular risk factor not a cardiovascular disease. This concept should be clarified.

Response 11: Thank you for the comment. We have clarified the differences between the concepts of cardiovascular disease and cardiovascular disease risk factors according to the previous studies and modified our manuscript accordingly (Page 6 , Line 183).

Point 12: Line 188: the authors discuss the relationship between body fat percentage and UA level in female staff. This finding has not been reported in the summarized key results (Line 164-169). It should be included.

Response 12: Thanks for the comment. To achieve a statistical power of 90% of our study, we combined male and female samples so that similar errors would not occur again.

Point 13: 169-170: The authors affirm that “These results indicated that the HPF test can be used as a predictor of UA, blood lipid, and GLU levels for overweight and obese individuals”. As the associations between HPF and Glucose is only significant for women, this affirmation should be modified because individuals are men and women.

Response 13: Thank you for the comment. To achieve a statistical power of 90% of our study, we combined male and female samples. When we combined the sample of men and women, the presentation of the results changed accordingly, so that the problems you mentioned were all resolved together.

Point 14: Line 241: Vital capacity is only significant in women. This fact should be mentioned

Response 14: Thanks for the comment. When we combined the sample of men and women, the presentation of the results changed accordingly, so that the problems you mentioned were all resolved together.

Point 15: Line 239: The authors affirm: “As one of the indicators for evaluating CRF, vital capacity…”. The concept of vital capacity as part of CRF should be reviewed.

Response 15: Thank you for the comment. In this part of the study, the results were no longer statistical significant as the male and female samples were combined. The misstatement in this section has been removed. Therefore the problems you mentioned were all resolved together.

Point 16: Limitations: It should be included that the study results cannot be extrapolated to the overall population because the sample only includes adults who were university staff.

Response 16: Thanks for the comment.  We have added the corresponding description in the limitation section to indicate that our results are not suitable for extrapolating to other populations (Page 7, Line 237).

Point 17: Line 82. Before this line, I recommend to add a section entitled Study Variables

Response 17: Thank you for the comment. We have added this section to make the research method more logical (Page 3, Line 92).

Point 18: Line 174: I suggest to Hypothesised with “may or might” before the verb “cause”

Response 18: Thanks for the comment. We have modified our manuscript accordingly (Page 6, Line 183).

Point 19: Line 189: I suggest to change the preposition “during”

Response 19: Thank you for the comment. We have replaced the preposition “during” with the more appropriate “in” (Page 5, Line176 ).

Round 2

Reviewer 1 Report

All previous comments were appropriately addressed. No further comments. Thank you.

Author Response

Thank you for your help. It is because of your comments that our manuscript has been improved.
Thanks again.

Reviewer 2 Report

Dear authors: you have considered the reviews I made and I think that the manuscript has been improved.

Author Response

Thank you very much for your great help in our submission.
Your carefulness, modesty and earnestness are our examples.
Thanks again.